# Alteration of the Fatty Acid Metabolism in the Rat Kidney Caused by the Injection of Serum from Patients with Collapsing Glomerulopathy

**DOI:** 10.3390/biomedicines8100388

**Published:** 2020-09-29

**Authors:** Elizabeth Soria-Castro, Verónica Guarner-Lans, María Elena Soto, María del Carmen Avila-Casado, Linaloe Manzano Pech, Israel Pérez-Torres

**Affiliations:** 1Cardiovascular Biomedicine, Instituto Nacional de Cardiología Ignacio Chávez, Juan Badiano 1, Col. Sección XVI, Mexico City 14080, Mexico; elizabethsoria824@gmail.com (E.S.-C.); loe_mana@hotmail.com (L.M.P.); 2Physiology Department, Instituto Nacional de Cardiología Ignacio Chávez, Juan Badiano 1, Sección XVI, Tlalpan, Mexico City 14080, Mexico; gualanv@yahoo.com; 3Immunology Department, Instituto Nacional de Cardiología Ignacio Chávez, Juan Badiano 1, Sección XVI, Tlalpan, Mexico City 14080, Mexico; mesoto50@hotmail.com; 4Pathology Department, University Health Network, Toronto General Hospital, 200 Elizabeth Street, Toronto, ON M5G 2C4, Canada; carmen.avila-casado@uhn.ca

**Keywords:** collapsing glomerulopathy, systolic blood pressure, fatty acids, proteinuria

## Abstract

Patients with collapsing glomerulopathy (CG) have marked proteinuria that rapidly progresses to chronic renal failure. In this study, we investigated if the nephropathy produced in a rat model by the injection of serum from CG patients induced alterations in fatty acid (FA) metabolism. Twenty-four female Sprague-Dawley rats were divided into four groups of six rats each: Group I, control rats (C); Group II, rats that received injections of 1 mL of 0.9% NaCl saline solution (SS); Group III, rats injected with 25 mg/mL of serum from healthy subjects (HS); and Group IV, rats injected with 25 mg/mL of serum from CG patients. In all groups, the systolic blood pressure (SBP), proteinuria, creatinine clearance (CC), cholesterol and total FA composition in the kidney and serum were evaluated. The administration of serum from CG patients to rats induced glomerular collapse, proteinuria, reduced CC and elevated SBP (*p* ≤ 0.01) in comparison with the C, SS and HS rats. The FA composition of the serum of rats that received the CG serum showed an increase in palmitic acid (PA) and a decrease in arachidonic acid (AA) when compared to serum from HS (*p* ≤ 0.02). In rats receiving the CG serum, there was also a decrease in the AA in the kidney but there was an increase in the PA in the serum and kidney (*p* ≤ 0.01). These results suggest that the administration of serum from CG patients to rats induces alterations in FA metabolism including changes in PA and in AA, which are precursors for the biosynthesis of the prostaglandins that are involved in the elevation of SBP and in renal injury. These changes may contribute to collapsing glomerulopathy disease.

## 1. Introduction

Collapsing glomerulopathy (CG) is a progressive and aggressive form of glomerular disease that was first described by Weiss et al. in 1986 [1], in a small group of six patients with nephrotic syndrome. Eight years later, Detwiler et al. [2], reported sixteen patients. CG has been described as a variant of focal and segmental glomerulosclerosis (FSGS) due to its morphological and physiological characteristics [3]. However, a recent investigation showed that the CG that is present in other nosologies has more aggressive characteristics than the rest of the FSGS types [4].

CG abnormalities include podocyte hyperplasia and hypertrophy, and these cells frequently present hyaline droplets. Lipid-laden macrophages of different sizes are present, and there is glomerular capillary collapse. In some cases, patients show bi-nucleate cells, tubular infiltration by macrophages, tubular dilation with an increase in the interstitial cell number, massive proteinuria over 10 g/24 h and rapid progression to terminal renal failure [5]. Some studies have described the presence of several circulating factors in the serum from CG patients such as the soluble urokinase plasminogen activator receptor (suPAR), cardiotrophin-like cytokine factor-1 (CLCF-1) and CD40 antibodies that may cause direct damage to podocytes, provoking their separation from the basal capillary membrane, which then results in the alteration of the permeability to albumin [6].

On the other hand, several in vitro and in vivo studies have suggested that abnormalities in fatty acid (FA) metabolism are present, which can participate in the modulation of renal damage, through glomerular and interstitial damage [7]. This phenomenon is observed from the initial stages of the disease to terminal renal failure [8]. There is also an association between renal injury and hyperlipidemia and obesity, which is related to alterations in FA metabolism, such as a decrease in polyunsaturated FAs (PUFAs) and an accumulation of saturated FAs (SFAs) [9]. However, it is not clear how these changes induce damage; alteration in the FA membrane composition can impair normal membrane function and result in renal injury [10]. Other studies have suggested that the formation of oxygenated products of arachidonic acid (AA) via cyclooxygenase (COX) and lipooxygenase (LOX) activities may be important in the progression of renal failure [11]. Abnormalities in the proportions of AA, which is one of the main precursors of vasoactive prostaglandins, may be implicated in the loss of the regulation of vascular tone, inducing high systemic blood pressure (SBP) and, eventually, renal injury [12].

In our laboratory, we have developed a Sprague-Dowley rat model that was described by Avila-Casado et al. in which an intravenous injection of serum from CG patients to the rats [13] induces specific characteristics that are commonly seen in CG such as proteinuria, decreased creatinine clearance (CC) and podocyte damage [5].

The purpose of this study was to evaluate if the nephropathy produced in the CG rat model, by the injection of serum from CG patients, was associated with FA metabolism abnormalities.

## 2. Materials and Methods

### 2.1. Obtainment of the Serum

The serum from 6 CG patients and 6 healthy subjects (HS) was obtained. The subjects agreed voluntarily to donate serum. The study was based on the ethical considerations of international ethical standards and the General Health Law, and on the Helsinki declaration, modified at the Congress of Tokyo, Japan [14]. The clinical and histopathological diagnosis of CG patients was performed by renal biopsy and by analyzing clinical history according to the Weiss criteria [1]. All patients were Mexican; seronegative for human immunodeficiency virus, parvovirus 19 and hepatitis C virus; and had no history of the use of intangible drugs. Medications that could interfere with the outcome of the study such as lipid-lowering drugs or non-steroidal anti-inflammatory drugs NSAIDs were suspended. The serum from the 6 HS was obtained from the blood bank and was matched by age and gender and used as a control. The serum from HS underwent a strict laboratory analysis to rule out other possible diseases. Blood was placed in tubes without anticoagulant. It was allowed to stand for a period of 10 min and was centrifuged at 644× *g* for 20 min at 4 °C. The serum was recovered and frozen at −30 °C. The Bradford method was utilized to determine the protein concentration in the serum [15]. A 25 mg amount of the protein from the serum of the CG patients or HS was diluted and adjusted to 1 mL with 0.9% NaCl solution (SS) and injected in the tail vein of Sprague-Dawley rats every 24 h, for 5 days.

### 2.2. Injections of Serum to Rats

The experiments on animals were approved by the Laboratory Animal Care Committee of the National Institute of Cardiology Ignacio Chávez and were conducted in compliance with the Guide for the Care and Use of Laboratory Animals of the National Institutes of Health NIH. Twenty-four female Sprague-Dawley rats weighing between 250 and 300 g and randomly distributed were used. They were divided into 4 groups of 6 rats each. Group I: control rats (C). Group II: rats that received injections of SS. Group III: rats injected with serum from healthy subjects (HS). Group IV: rats injected with serum from CG patients (CG). Before the injection, the animals were anesthetized with ether. They were put into individual metabolic cages (Nalgene, New York, USA) with free access to water and food. All the animals received commercial food for rodents that contained 23% crude protein, 4.5% crude fat, 8% ash and 2.5% added minerals (Lab Diet 5008; Richmond, IN, USA).

### 2.3. Systolic Blood Pressure and FA Determinations

At the end of the treatment, the SBP was measured by the tail-cuff method [16]. Five to six measurements were performed on each animal. Urine was filtered and collected at 24 (baseline), 48, 72, 96 and 120 h. Twenty-four hours after the last injection and after overnight fasting, the rats were anaesthetized with sodium pentobarbital (60 mg/kg, Pfizer, Mexico), and blood was collected from the abdominal aorta using a syringe. The blood was centrifuged at 644× *g* for 20 min at 4 °C. The serum was isolated and kept with 0.02% of butylated hydroxytoluene BHT as an antioxidant at −30 °C.

The kidneys were perfused in situ with 0.9% NaCl, dissected, decapsulated and homogenized with buffer containing 0.25 M sucrose, 10 mM Tris, 1 mM EDTA (pH = 7.35) and protease inhibitors (1 mmol phenylmethylsulfonyl fluoride** PMSF, 2 mM leupeptine, 2 mM pepstatine A, and 0.1% aprotinine A (*w*/*v*)). The kidneys were homogenized, and the homogenate was centrifuged at 447× *g* for 10 min at 4 °C. The supernatant was separated and kept with 0.02% BHT at −30 °C. For the analysis of FA composition, 0.1 mL of serum or 100 mg of protein from the homogenized kidney was used as described by Folch et al. [17]. FA methyl esters were separated and identified by gas chromatography–Flame Ionization Detector FID in a Carlo Erba Fratovap 2300 chromatograph equipped with a capillary column packed with the SP-2330 phase (30 m long and with a 0.25–0.2 mm film thickness) and fitted with a flame ionization detector at 210 °C, with helium as the carrier gas at a flow rate of 1.2 mL/min. Cholesterol (CT) was quantified in the serum and kidney homogenate. In brief, 0.1 mL of serum or 100 mg of protein from kidney homogenate was added to 30 µg of stigmasterol as an internal standard and 1 mL of 0.1 N KOH. The mixture underwent agitation for 30 s and was heated for 30 min at 30 °C. At the end of the reaction, 1 mL of NaCl 0.09% and 3 mL of anhydrous ether were added, mixed vigorously for 20 s and centrifuged at 447× *g* for 5 min. The ether phase was separated and dried over anhydrous sodium sulfate to eliminate any traces of water. The recovered ether phase was evaporated under nitrogen gas. The sample obtained was dried under vacuum conditions in the presence of diphosphorus penta-oxide (P_2_O_5_) overnight. Then, 100 µL of pyridine, 50 µL of hexamethyldisilazane and 30 µL of trimethylchlorosilane were added to the obtained residue. The mixture was agitated vigorously and heated at 60 °C for 30 min. At the end of the reaction, the excesses of the reagents were evaporated under nitrogen gas. The formed CT derivatives were extracted with 2 mL of hexane and filtered and evaporated under nitrogen. The hydroxy-trimethyl-silanes of CT obtained were quantified by gas chromatography–FID.

Albuminuria was measured using the bromocresol green reagent. This technique is specific for the quantification of albumin in urine [18]. Serum and urine creatinine were measured by the Jaffe method [19], and glomerular filtration was calculated according to the following formula: [UCr]/[SCr]*Urinary volume/Time (UCr: urinary creatinine, SCr: serum creatinine). The FA composition of the serum from the CG patients and HS was analyzed following the method mentioned above (Table 1).

### 2.4. Histopathological Analysis

Tissue was processed for light microscopy according to standard techniques. The kidney was dissected, decapsulated and washed in 0.9% NaCl for 30 s, fixed in 10% formalin solution for 24 h, gradually dehydrated in ethanol, cleared in xylene and embedded in paraffin. The kidney was cut into five-micrometer-thick slices with a microtome (Leica RM212RT, Wetzlar, Germany); the paraffin sections were stained with Masson trichrome stain, periodic acid–Schiff stain (PAS) and Jones’ methenamine silver stain. For immunohistochemical staining, other sections were mounted on slides treated with poly-lysine, deparaffinized, rehydrated and treated with 0.1 M citrate buffer to unmask antigens in a pressure cooker. The slides were mounted on the cover plates, and the technique was carried out in a slide rack. It was incubated with primary monoclonal antibodies against TNFα (H-156) (Santa-Cruz Biotechnology, Dallas, TX, USA), WT1 (6F-H2) (Diagnostic BioSystems, San Diego, CA, USA) and 5-LOX (33) (Santa-Cruz Biotechnology) at final dilutions of 1:50 for 24 h at 4 °C. Then, the samples were washed with Tris buffered saline TBS three times, incubated for 45 min at room temperature with MACH2 Rabbit or MACH2 mouse HRP-Polymer (Biocare Medical, Concord, CA, USA), treated with DAB (3′3′-diaminobenzidine), contrasted with Hematoxylin Hill’s and mounted for observation and analysis. Averages of 20 glomeruli per level of section in each sample were used. Histological sections were analyzed using a light microscope, Carl Zeiss (63300 model), equipped with a Tucsen (9 megapixels) digital camera with the software TSview 7.3.1 at a 400× magnification. The photomicrographs were analyzed by densitometry using the Sigma Scan Pro 5 Image Analysis software, Systat Software Inc., San Jose, CA, USA. The density values are expressed as pixel units.

Electron microscopy examination was performed in all cases. The tissue was sectioned into pieces of 1 mm of thickness, fixed by immersion in 2.5% glutaraldehyde, post-fixed in 1% OsO_4_ buffer, dehydrated with increasing concentrations of ethanol and infiltrated with epoxy resins. Ultrathin sections of 60 nm were contrasted in uranyl acetate and lead citrate to be further examined with a JEM-1011 (JEOL Ltd., Tokyo, Japan) at 60 Kv. Random pictures were taken. Three to five glomeruli of each rat were examined at a 12,000× magnification.

### 2.5. Statistical Analysis

Statistical analysis was performed using the Sigma Plot program version 14 (Sigma Plot, Jandel Corporation, San José, CA USA, 1986–2017). Statistical significance was determined between two groups by Mann–Whitney U tests and among all groups by one-way ANOVA tests, followed by post-hoc Tukey tests. Differences were considered statistically significant when *p* ≤ 0.05. The data are presented as mean ± standard deviation.

## 3. Results

Table 1 shows the general characteristics of the 6 CG patients. 4 were male and 2 were female. There were no significant differences in CC, blood pressure, urinary albuminuria, CT, triglycerides, uric acid blood urea nitrogen and collapsed glomeruli.

Comparison of FA composition of serum from CG patients and HS showed that there were significant statistical differences in SFA such as palmitic acid (PA), which was higher in the serum of CG patients compared with the serum from HS (*p* = 0.01). Regarding the PUFA, the AA and docosahexaenoic acid were significantly decreased in the serum from CG patients when compared with the serum from HS (*p* = 0.001 and *p* = 0.01 respectively). The total SFA in the serum from CG patients were significantly increased (*p* = 0.001) in comparison with HS, while the PUFA (n-3) in the serum from CG patients were lower (*p* = 0.04, Table 2).

Table 3 shows the general characteristics of the rat experimental groups during the experimental period. No differences in the amount of drinking water and in the food intake between the groups with or without treatment were found. Each rat consumed, daily, 22.80 ± 3.09 g of food and 40 ± 10 mL of water. However, urinary protein and SBP were significantly increased, while CC was decreased, in the CG group in comparison with the C, SS and HS groups (*p* < 0.001).

Table 4 shows the analysis of the FA composition of the kidney homogenates from the experimental groups of rats. In the CG group, a statistically significant increase in the proportions of PA and palmitoleic acid in comparison with the C (*p* = 0.04 and *p* = 0.01), SS (*p* = 0.04) and HS (*p* = 0.01 and *p* = 0.04) groups was observed. The proportions of dihomo-γ-linolenic acid and AA were increased and reduced, respectively, in the CG group when compared with the C (*p* = 0.04 and *p* = 0.001), SS (*p* = 0.04 and *p* = 0.01) and HS (*p* = 0.005) groups. The increased PA and palmitoleic acid proportions resulted in a statistically significant increase in the total SFAs and in the total MUFAs in the CG group (*p* = 0.04) in comparison with all of the other groups. The PUFAs (n-3) were reduced due to the decrease observed in the AA in the CG group in comparison with C (*p* = 0.001), SS (*p* = 0.01) and HS (*p* = 0.005).

The ratios of C16:1n-7/C16:0 and C20:4n-6/C20:3n-6, usually used as indexes of the ∆^9^- and ∆^5^- desaturation activities for PA and dihomo-γ-linolenic acid, showed a significant increase in the CG group when compared with the C (*p* = 0.04 and *p* = 0.01), SS (*p* = 0.05 and *p* = 0.01) and HS (*p* = 0.04 and *p* = 0.01) groups, respectively (Table 5).

Table 6 shows the FA compositions of the serum from the experimental animals. A significant increase was observed in the PA and oleic (OA) acid proportions in the serum from the CG group in comparison with the C (*p* = 0.05), SS (*p* = 0.04) and HS (*p* = 0.04 and *p* = 0.01) groups. Linoleic acid showed a significant decrease in the CG group in comparison with the C (*p* = 0.05) and SS (*p* = 0.04) groups. γ-linoleic acid showed a significant decrease in the CG group in comparison with the C (*p* = 0.01), SS (*p* = 0.01) and HS (*p* = 0.05) groups. α-linoleic acid showed a significant decrease in the CG group in comparison with the C (*p* = 0.001) group. A significant increase in the total MUFAs was noted in the serum from rats treated with the serum from CG patients in comparison with the C (*p* = 0.05) and HS (*p* = 0.001) groups.

The ratios of C16:1n-7/C16:0, C18:1n-9/C18:0 and C20:4n-6/C20:3n-6 were significantly decreased, while the ratio of C18:3n-6/C18:2n-6 and was significantly increased, in the serum of the C (*p* = 0.03), SS (*p* = 0.04 and *p* = 0.01) and HS (*p* = 0.05, *p* = 0.01) groups, respectively, in comparison with the CG group (Table 7).

### 3.1. Histology

Figure 1, Figure 2, Figure 3 and Figure 4 show representative photomicrographs of the renal cortexes from the kidneys of the experimental groups. The glomeruli showed open capillary loops, basement membranes and Bowman’s capsules positive for PAS staining in the C, SS and HS groups. (Figure 1, Panels A–C), and Masson and Jones’ methenamine staining in the C, SS and HS groups (Figure 3, Panels A–C, and Figure 4, Panels A–C, respectively). However, in the rats injected with the serum from CG patients, the glomeruli showed podocyte hypertrophy, a reduced glomerular size, retraction of the tuft and prominent visceral cells. There was an increase in the urinary space, with some vacuoles in it and with and without immunolabeling for TNFα and WT1 (Figure 5 and Figure 6), respectively. These changes are possibly a precursory finding for glomerular collapse; however, glomerular loop collapse was not clearly demonstrated. Probably, a longer time of evolution would be needed to observe glomerular collapse (Panel D).

The glomeruli from rats that received serum from patients with CG show a diffuse glomerular tuft retraction. The tubules do not show alterations in the first three groups, but in the CG group, the PAS staining is positive in the proximal convoluted tubules, and there are vacuoles and edema (Figure 2).

The densitometric analysis of the glomerular area showed a significant decrease in the CG group in comparison to the C (*p* = 0.001), SS (*p* = 0.009) and HS (*p* = 0.001) groups; Figure 5.

### 3.2. Immunohistochemistry

Figure 6 and Figure 7 show the representative immunolabeling for WT1 and the densitometric analysis in the glomeruli. The densitometric analysis of the immunolabeling showed a decrease for WT1 in the C (*p* = 0.03), SS (*p* = 0.007) and HS (*p* = 0.04) groups in comparison with the CG group.

Figure 8 and Figure 9 show the representative immunolabeling and densitometric analysis for TNFα in the glomeruli, respectively. The densitometric analysis of the immunolabeling showed a decrease for TNFα in the C (*p* = 0.001), SS (*p* = 0.006) and HS (*p* = 0.001) groups in comparison with the CG group.

Figure 10 and Figure 11 show the representative immunolabeling and densitometric analysis for 5-LOX in the glomeruli, respectively. The densitometric analysis of the immunolabeling showed a decrease for 5-LOX in the C (*p* = 0.02) and HS (*p* = 0.002) groups in comparison with the CG group.

Figure 12 shows representative photomicrographs of the ultrastructure of a rat glomerulus. No abnormalities were observed in C, SS and HS. However, in the CG rats, there was a collapse of the glomerular membranes and extensive obliteration processes.

## 4. Discussion

The purpose of this study was to investigate the mechanism by which the intravenous administration of serum from CG patients into rats induces high SBP by modifying the FA profile in the kidney and serum in rats. These changes may contribute to the glomerular collapse and the decrease in the CC. Several studies have indicated that microproteinuria is a marker of glomerular damage and that it predicts the development of overt proteinuria and progressive renal failure [20,21]. In experimental models and patients with essential hypertension, the combined presence of microproteinuria and hyperlipidemia is frequent [22], and this may cause renal damage that results in an increase in urinary microproteinuria [7]. The results show that CG patients developed high blood pressure, proteinuria and dyslipidemia. Both hypercholesterolemia and hypertriglyceridemia contribute to hyperlipidemia, and this may contribute to mesangial cell proliferation and cause endothelial dysfunction and the neutralization of glomerular basement membrane anionic sites that leads to the progression of the nephrotic syndrome. However, the exact mechanism is not completely understood [23]. In our model, there was a presence of albuminuria and increased SBP in the CG rat group. The CT levels in the kidney showed a tendency to increase without reaching a statistically significant change. These results suggest that the serum of CG patients contains circulating permeability factors that contribute to impaired renal hemodynamics, which are reflected in massive proteinuria and increased SBP [12]. The factor or factors present in the serum from CG patients implicated in the induction of these abnormalities are not known. However, recent investigations have described three possible candidates: suPAR; CLCF-1, a member of the IL-6 family of cytokines; and CD40 antibodies such as a potential permeability factor. The three molecules can lead to CG abnormalities, but to date, different studies also have questioned their validity [24].

The results from the histopathological studies showed that the CG serum administration in rats led to nephrotic syndrome, which was characterized by a glomerular tuft retraction, the collapse of the glomerular loops, hypertrophy, increased prominent visceral cells and the presence of some vacuoles in the urinary space. The glomeruli from rats that were injected with the serum from patients with CG showed a retraction of the glomerular tuft and droplets within visceral epithelial cells. However, no marked glomerular collapse was observed. PAS staining was positive in the proximal convoluted tubules and in the basal lamina of the tubules and glomeruli. The tubules showed no changes or alterations in any of the other experimental groups. Although there was not significant damage to the tubules, eosinophilia was observed in the proximal tubules, especially in those surrounding glomeruli with tuft retraction. This may be the result of tubular damage from proteinuria, as albumin droplets can be produced. No reticulofibrillar structures were found in the endothelium of the capillary loops by electron microscopy in any of the cases.

Furthermore, the immunohistochemical results showed that the CG group was positive for TNFα. CG is also characterized by high levels of pro-inflammatory processes, where TNFα plays an important role [25]. However, our results showed negativity for WT1, a marker of the podocyte maturation that disappears in CG. The changes observed in this study are possibly a precursor of a future glomerular collapse finding. It is possible that a longer evolution time would be needed to observe true glomerular collapse. However, the alteration in the FA profile may contribute to this evolution.

On the other hand, abnormalities in FA metabolism can contribute to the modulation of the renal damage, through interstitial and glomerular injury, from the initial stages of the disease to terminal renal failure [8]. There is also a narrow association between renal injury and hyperlipidemia and obesity, which are related to an altered FA metabolism where there is a decrease in polyunsaturated FAs (PUFAs) and accumulation of saturated FAs (SFAs) [9]. However, to date, the exact metabolic pathway by which the alteration in the FA profile may induce renal damage is unknown. Probably, the alteration in the FA profile leads to modifications in the permeability of the bilipid cellular membrane, which may impair normal membrane function, including changes in metabolite exchange and in the activity of membrane-bound enzymes, receptors and humoral transduction that results in renal damage [10]. In this sense, the total FA composition of the lipid fraction extracted from the serum of the CG patients and HS was analyzed to investigate if the changes observed in the FA composition in the rat serum and kidney homogenate could be a reflection of the changes in the original serum or if they were associated with the FA composition of the serum from the patients, thus contributing to the kidney alteration. An increase in PA but a decrease in AA and γ-linolenic and docosahexaenoic acids was observed in the serum from the CG patients when compared with the serum from HS. However, no change in the proportion of AA was noted in the serum from the CG rat group, while γ-linolenic, α-linolenic and dihomo-γ-linolenic acid were reduced in this group. These results suggest that the administration of serum from CG patients who have dyslipidemia to animals induces an alteration in the metabolism of PUFAs, including precursors of AA biosynthesis, since these changes did not occur in the percentages of those FAs in the rats that received serum from HS or SS. Furthermore, an increase in the percentage of OA in the serum from CG rats was observed. This increase cannot be attributed to a lower food intake, because there was no difference in the consumption of diet-chow between the CG and C, and SS and HS groups. The change may result from the increase in ∆^9^-desaturase activity as indicated by our results. The activity of this enzyme can be calculated indirectly from the rate of conversion resulting from the addition of the product divided by the substrate from its predecessor (palmitoleic/palmitic or oleic/stearic). The desaturation process is important for the regulation of the integral proteins in the cell membrane, and of fluidity, and may contribute to different pathological disorders including nephrotic syndrome [26]. Our results suggest that the desaturation process performed by the ∆^9^-desaturase is altered in CG, which can contribute to the modification of the FA profile with an increase in PA and palmitoleic acid, which participate in the increase and decrease in the rigidity and fluidity of the membranes of podocytes and mesangial cells. In this sense, renal lipid accumulation can cause structural and functional changes in mesangial cells, podocytes and proximal tubule cells, which contributes to the impairment of the function of the nephron [27].

In addition, one of the main sites of renal lipid accumulation is the renal proximal tubule. High levels of albumin-bound long-chain SFAs may promote the progression of renal tubular damage and interstitial fibrosis, coupled with fibrosis in the interstitium of the tubule. The high lipid accumulation can also contribute to the development of glomerulosclerosis [28]. Moreover, in patients with obesity that is associated with renal damage and high proteinuria, the renal biopsies show glomerular hypertrophy and FSGS lesions. The glomerulosclerosis induced by lipid accumulation may be the result of the concerted activation of sterol regulatory element binding proteins. These proteins are indirectly required for the uptake and biosynthesis of FAs and CT [28].

In addition, OA is one of the FAs more frequently increased in the serum from humans with essential hypertension and dyslipidemia, in experimentally hypertensive animals, and in animals under high-fat diets [29]. OA may contribute to the pathogenesis of endothelial dysfunction by releasing cytokines including IL-6, by increasing TNFα production, and by inducing apoptosis, necrosis and oxidative stress [30]. It has also been described that the treatment of rats with a high-fat diet leads to chronic inflammation and the development of glomerular damage, through the accumulation of FAs that can promote the overexpression of CD36, TNFα, IL-6 and monocyte chemotactic protein-1 during inflammation. Inflammation, in turn, results in the thickening of the glomerular basement membrane, increased extracellular matrix and glomerulosclerosis [31]. Furthermore, in HK-2 cells, the accumulation of lipids can result in endoplasmic reticulum stress and an increase in TNFα and IL-6, resulting in the elevated production of reactive oxygen species with direct toxic effects on the kidneys [31].

Indeed, γ- and α-linolenic acids decrease blood pressure in spontaneously hypertensive rats. The PUFA n-6 essential FA lowers blood pressure in hypertensive humans [32]. However, our results show that a decrease in this PUFA can contribute to increase SBP. In addition, the lower proportions of AA and dihomo-γ-linolenic acid in the kidney could be due to the alteration of ∆^5^-desaturation activity, which is a limiting-step in the biosynthesis of PUFAs [33]. An alteration of AA biosynthesis could be a determining factor increasing the synthesis of prostaglandins such as PGE_2_ and TXA_2_. These molecules are involved in the regulation of vascular tone and in the inflammatory process, where TNFα participates [9,12]. Thus, an imbalance in the distribution of the C20-PUFAs (dihomo-γ-linolenic acid, AA and eicosapentaenoic acid) could have contributed to the development and the maintenance of hypertension and impaired kidney function in the CG group. It is possible that an alteration in tissue FAs is directly associated with renal injury. Indeed, changes in essential FA metabolism may stimulate cell growth and proliferation, and the release of cytokines and inflammatory processes, and may have an important effect possibly mediated by altered eicosanoid production [33,34]. It has been suggested that LOX and the cyclooxygenase products of AA metabolism are important in the inflammatory process in progressive renal injury [12,33,35]. The LOX pathway oxidizes AA to one or more hydroperoxyeicosatetraenoic acids (hydroperoxy-HPETEs) that are then reduced to the hydroxyeicosatetraenoic acids (HETEs). However, HPETEs can be converted either to their respective hydroxyl FAs or to leukotrienes (LTs). LTs are compounds with vasoconstrictor actions [36]. Our results showed positivity for 5-LOX in the glomeruli of the rats that were injected with the serum from patients with CG. This suggest that the AA pathway through to the 5-LOX pathway has an important role in the CG increase in vasoconstriction, which decreases the blood flow and decreases the CC. This results in edema, which is a feature present in patients with CG.

In addition, the metabolites of the 5-LOX pathway—LTB4, LTC4, LTD4 and LTE4—participate as pro-inflammatory and vasoconstrictor agents, they cause leukocyte adherence to the vascular endothelium, and they participate in the vascular remodeling and proliferation of muscular cells. In the kidney, they may also regulate glomerular circulation by regulating prostacyclin production and participate in glomerular disease [11]. Therefore, they could participate in glomerular collapse in this pathology. However, more studies are required to corroborate this hypothesis.

In FSGS, several experimental agents can inhibit the albumin oncotic pressure in the glomeruli, and this can lead to a decrease in this disease. Among the possible molecules involved are eicosanoids such as 20-hydroxyeicosatetraenoic and 8, 9-epoxyeicosatrienoic acids, which are metabolites of AA’s metabolism via cytochrome p450 or by the inhibition of the cyclooxygenase [35].

The present investigation of serum-induced CG in rats showed a relationship between alterations in FA metabolism and renal injury and hypertension. An association between renal injury and alteration in lipid metabolism was also previously found in obese Zucker rats, which is a model of endogenous hyperlipidemia and spontaneous renal injury [37,38]. Other investigators have found increased renal injury in experimental animals fed a diet rich in CT [7,8]. The CT in the diet induces changes in the metabolism of FAs such as an increase in the MUFAs and a decrease in the PUFAs including AA, by the inhibition of ∆^5^-desaturase activity [33,39]. An increase in the cortical CT and an alteration in the renal FA profile reflect the deficiency in essential PUFAs, which is associated with glomerular and tubulointerstitial damage [40]. In this study, the injection of serum from CG patients who had an increased concentration of CT did not alter the concentration of CT in the serum and in the kidney homogenates of the rats. These results suggest that the decrease in the AA-to-dihomo-γ-linolenic acid ratio, an index of ∆^5^-desaturase activity, and the decrease in the AA proportion in the kidney homogenate may be due to the high level of serum CT from the CG patients injected into the rats. CT, when administered in the diet, induces a decline in the proportion of AA [33,41]. It has been shown that the excess lipid accumulation in the renal parenchyma is relevant to the development of chronic kidney disease (CKD) and can extend the damage at the tubular and glomerular levels. Therefore, there are sufficient bases for proposing the reduction of circulating lipid levels in the treatment of these patients.

There are several studies that support that improving lifestyle may reduce damage. Weight loss promotes the anti-proteinuria effect of angiotensin II receptor blockers in patients with CKD [42]. Large studies where soy and isoflavone consumption has been included in the diet significantly reduced total CT, LDL-CT, serum triglycerides, serum C-reactive protein, proteinuria and urinary creatinine levels [43]. There are other dietary approaches to stopping hypertension such as the ingestion of a Mediterranean diet, among others. These diets control weight and prevent hypertension, diabetes and urinary albumin, thus improving kidney function and reducing the risk of kidney damage [44,45]. The n-3 PUFAs in the diet are involved in the regulation of the immune system, the inflammatory and metabolic pathways induced by several substances, signal transduction and cell membrane formation. In addition, they reduce blood pressure and triglycerides, and they can be of great benefit in CKD [46,47]. In CKD patients, diet and lifestyle control, as well as the timely use of lipid-lowering drugs, are factors that should be considered as useful therapeutic interventions and should be evaluated through randomized clinical trials to determine the effects of nutritional status on kidney damage and lipid metabolism in CG patients.

## 5. Conclusions

Since lipid metabolism is one of the most important physiological processes, our results show the far-reaching involvement of the alteration of FA metabolism in the renal injury associated with elevated SBP. The major cause of the end-stage renal failure is the elevated renal blood pressure, which can be due to the excessive production of vasoconstrictors and decrease in vasodilators due the alterations in the proportion of AA. This study provides information on the possible mechanisms implicated in the renal injury and in the elevation of blood pressure due to the administration of serum from CG patients, based on the possible alterations in the metabolism of AA. Nevertheless, the factor or factors present in the serum from CG patients and responsible for these alterations remain unknown. To elucidate the nature of these factors, further experiments will be undertaken in our laboratory. On the other hand, the precise mechanisms responsible for the deleterious effects of lipids on glomerular function are not well established and need further investigation.

### Limitations of the Study

The main limitation of this study is the absence of a rat group injected with serum from other pathologies that involve proteinuria and kidney damage, such as other FSGS that are different to CG. These pathologies may also induce changes in the FA profile.

## Figures and Tables

**Figure 1 biomedicines-08-00388-f001:**
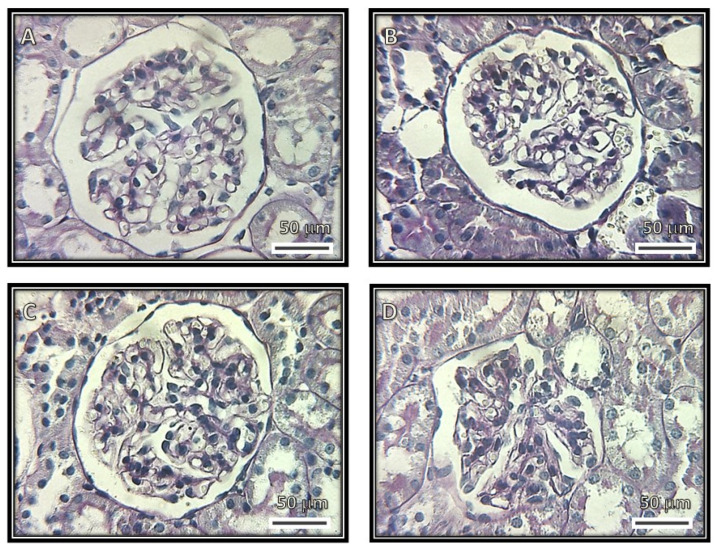
Representative photomicrographs of rat renal cortexes. The images show glomeruli from (**A**) control, (**B**) kidneys from rats injected with saline solution and (**C**) kidneys from rats injected with serum from healthy subjects. Open capillary loops, basement membranes and Bowman’s capsules positive for periodic acid–Schiff stain (PAS) staining can be observed in these panels. (**D**) Kidneys from rats that were injected with serum from patients with collapsing glomerulopathy. Retraction of the glomerular tuft, podocyte hypertrophy and injury, increased urinary space and presence of some vacuoles in the urinary space can be observed. (Periodic acid–Schiff stains, 400×.)

**Figure 2 biomedicines-08-00388-f002:**
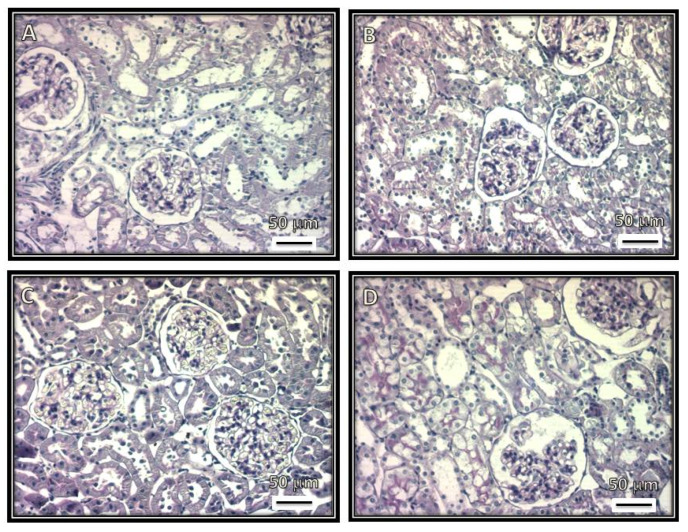
Representative photomicrographs of rat renal cortexes. (**A**) Control. (**B**) Kidneys from rats injected with saline solution, and (**C**) rats receiving serum from healthy subjects. Open capillary loops, basement membranes and Bowman’s capsule positive for PAS staining are observed in these panels. (**D**) Kidneys from rats receiving serum from patients with CG. Retraction of the glomerular tuft, podocyte hypertrophy and injury and increased urinary space are observed, and there are some vacuoles in the urinary space. (Periodic acid–Schiff stains, 200×.)

**Figure 3 biomedicines-08-00388-f003:**
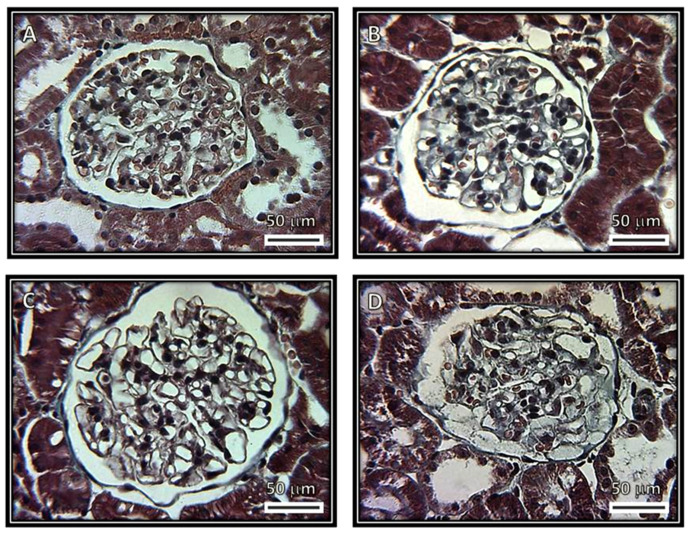
Representative photomicrographs of rat renal cortexes. Images show glomeruli from (**A**) control, (**B**) kidneys from rats injected with saline solution and (**C**) rats that received serum from healthy subjects. No abnormalities were observed under light microscopy in these groups. There are open capillary loops and conserved spaces. (**D**) Rats that received serum from patients having collapsing glomerulopathy. Glomeruli show podocyte hypertrophy, retraction of capillary loops and prominent visceral cells (Masson staining, 400×).

**Figure 4 biomedicines-08-00388-f004:**
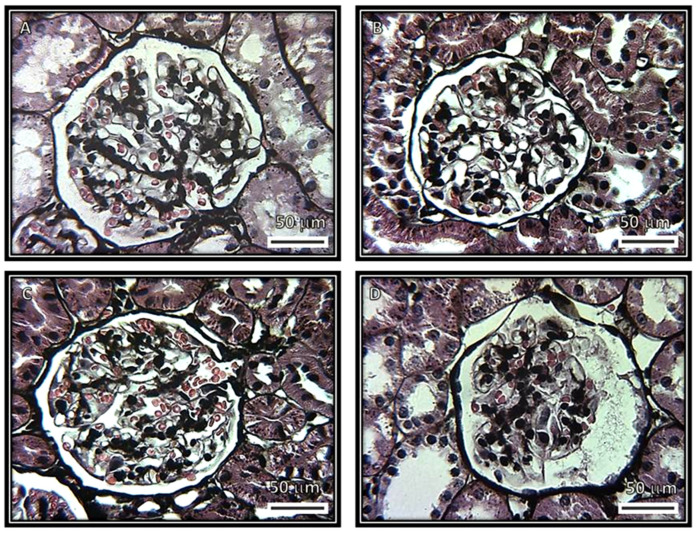
Representative photomicrographs of rat renal cortexes. Images show glomeruli from (**A**) control, (**B**) kidneys from rats that were injected with saline solution and (**C**) kidneys from rats that received serum from healthy subjects. No abnormalities were observed under light microscopy in these groups. There are open capillary loops and conserved spaces. (**D**) Cortexes from rats that received serum from patients having collapsing glomerulopathy. Glomeruli show podocyte hypertrophy, collapse and retraction of the glomerular tuft (400× Jones’ methenamine silver reaction). There are a collapse of capillary loops and prominent visceral cells.

**Figure 5 biomedicines-08-00388-f005:**
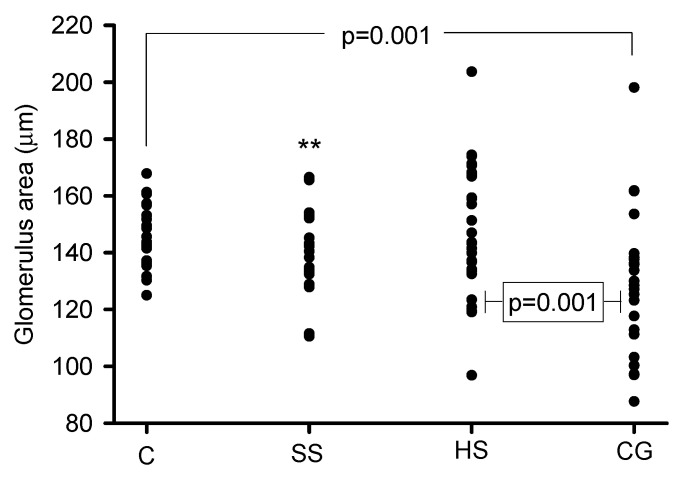
Densitometric analysis of the glomerular area in the experimental groups. The collapse of the glomerulus was evident in CG group in comparison to the C, SS and HS groups; ** SS vs. CG, *p* = 0.009.

**Figure 6 biomedicines-08-00388-f006:**
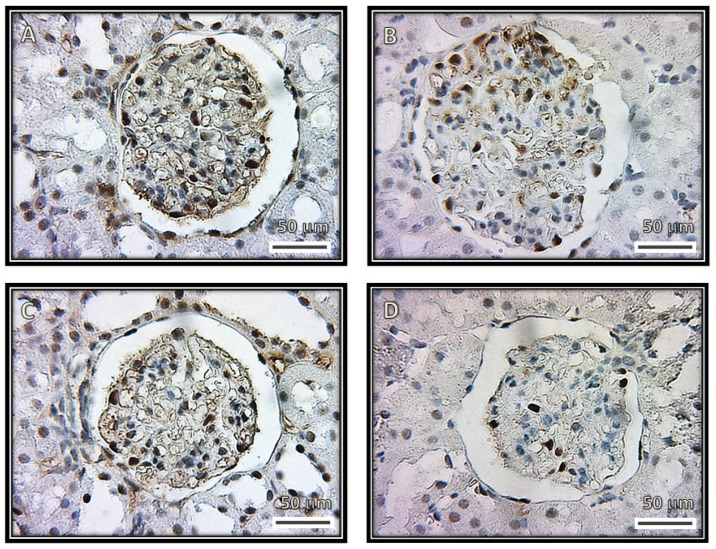
Representative immunohistochemistry of rat renal cortex with the ripening marker WT1. The images show glomeruli from (**A**) control, (**B**) kidneys from rats that were injected with saline solution and (**C**) kidneys from rats that received serum from healthy subjects. The images in these three panels are positive for visceral cells. (**D**) Kidneys from rats that received serum from patients having collapsing glomerulopathy. The image is negative for visceral cells.

**Figure 7 biomedicines-08-00388-f007:**
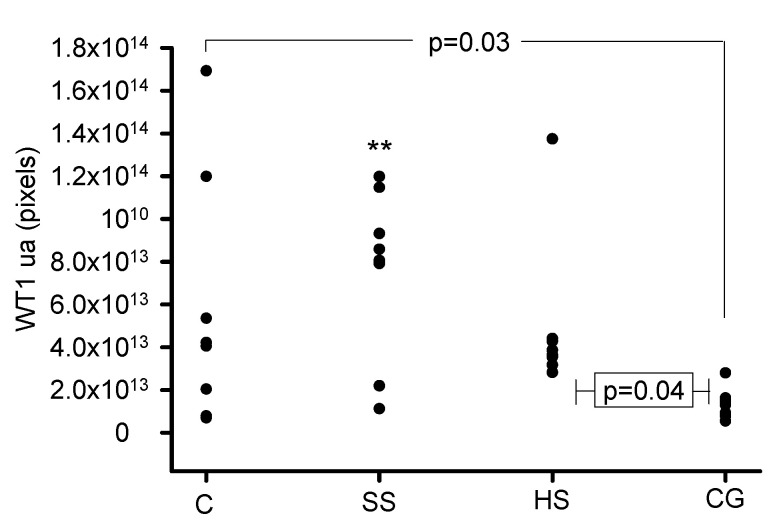
Densitometric analysis of the glomerulus for the ripening marker WT1 in the experimental groups. The labeling was positive in the C, SS and HS groups and negative in the CG group. ** SS vs. CG, *p* = 0.007. Each point is the mean of the analysis of 5 glomeruli per group.

**Figure 8 biomedicines-08-00388-f008:**
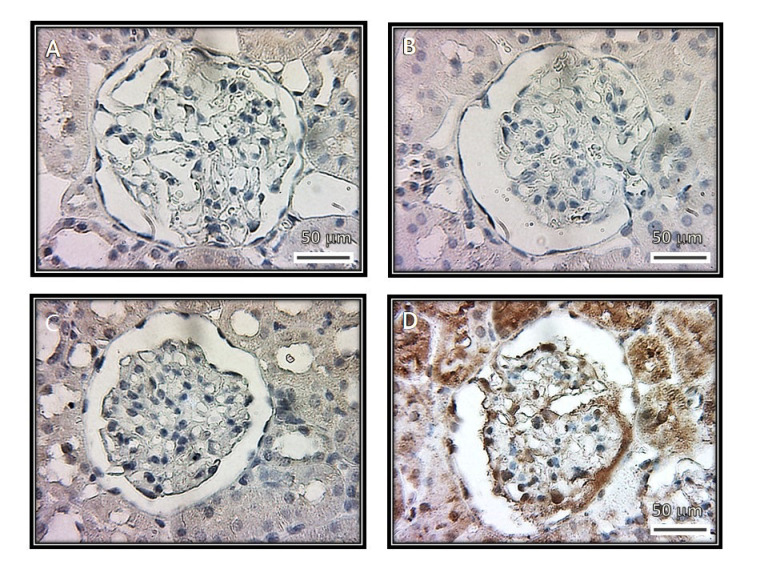
Immunohistochemistry of representative rat renal cortexes with the TNFα antibody. The images show glomeruli from (**A**) control, (**B**) kidneys from rats that were injected with saline solution and (**C**) kidneys from rats that received serum from healthy subjects. The images in these three panels show a negative stain. (**D**) Kidneys from rats that received serum from patients having collapsing glomerulopathy. In this panel, there is a positive stain.

**Figure 9 biomedicines-08-00388-f009:**
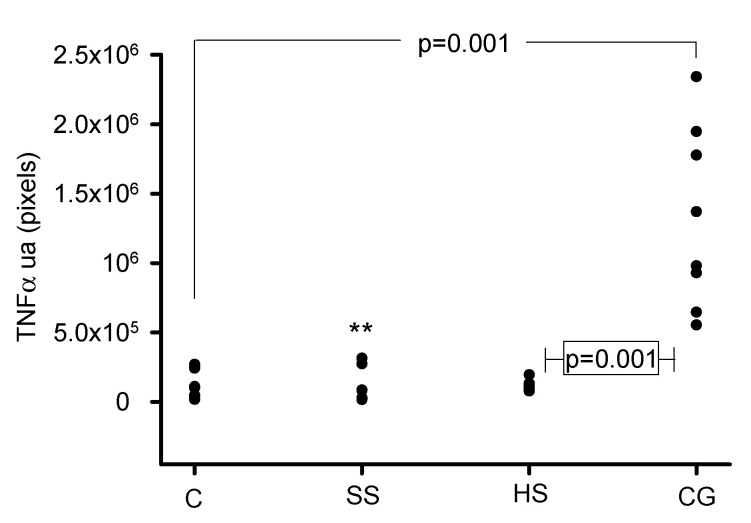
Densitometric analysis for the marker TNFα in the glomeruli from the experimental groups. The TNFα antibody was negative for the C, SS and HS vs. CG groups; ** SS vs.CG, *p* = 0.006. Abbreviations: C = control, SS = saline solution, HS = healthy subjects and CG = collapsing glomerulopathy (*n* = 6). Each point is the mean of the analysis of 5 glomeruli per group.

**Figure 10 biomedicines-08-00388-f010:**
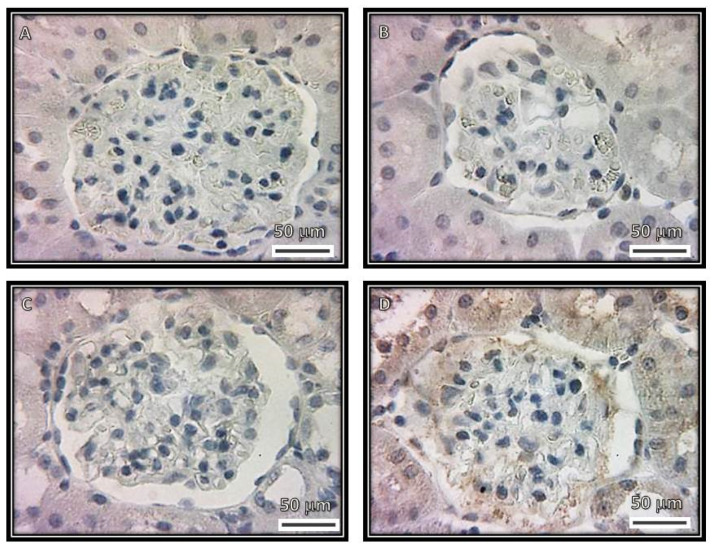
Immunohistochemistry of representative rat renal cortexes for the 5-LOX antibody. The images show glomeruli from (**A**) control, (**B**) kidneys from rats that were injected with saline solution and (**C**) kidneys from rats that received serum from healthy subjects. The images in these three panels show negative stains. (**D**) Kidneys from rats that received serum from patients having collapsing glomerulopathy. In this panel, there is a positive stain.

**Figure 11 biomedicines-08-00388-f011:**
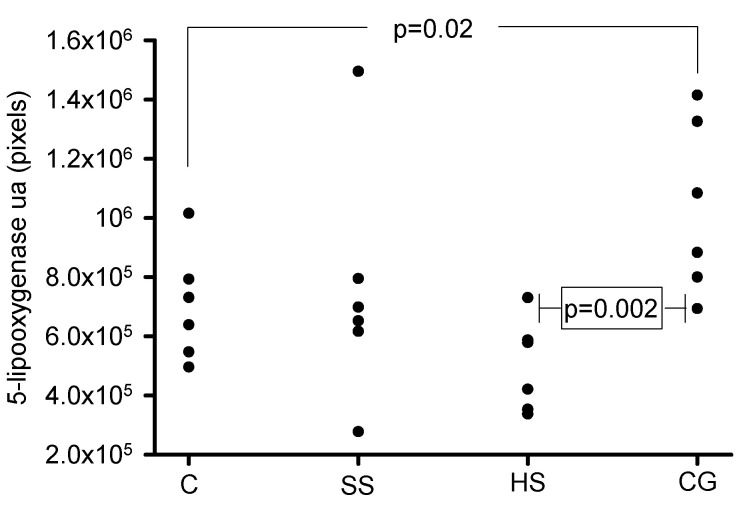
Densitometric analysis for the marker 5-LOX in the glomeruli from the experimental groups. The 5-LOX antibody was negative for the C, SS and HS vs. CG groups. Abbreviations: C = control, SS = saline solution, HS = healthy subjects and CG = collapsing glomerulopathy (*n* = 6). Each point is the mean of the analysis of 5 glomeruli per group.

**Figure 12 biomedicines-08-00388-f012:**
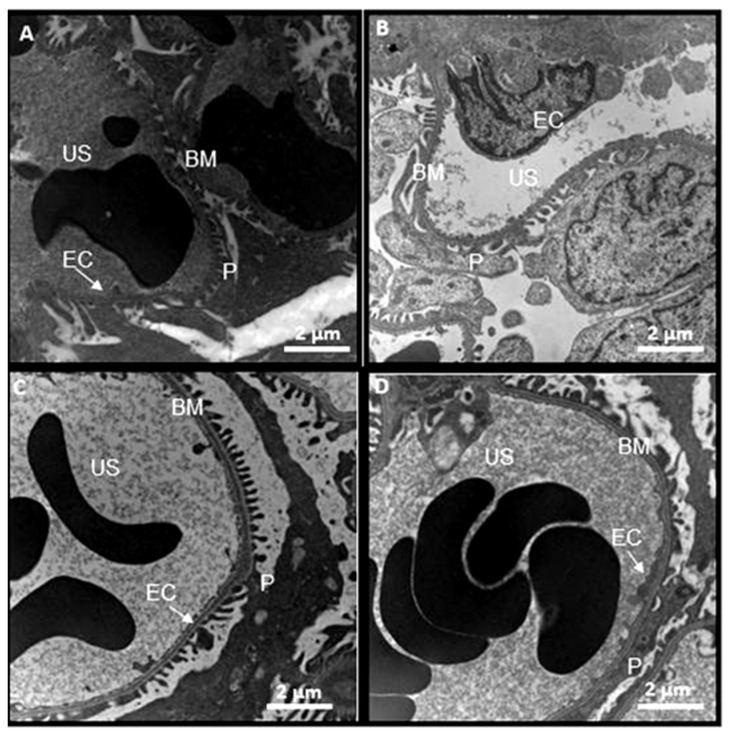
Electron photomicrographs representative of the capillary loops and the ultrastructure of rat glomeruli from the experimental groups at 12,000×. The collapsing glomerulopathy group showed podocyte foot processes and a display of abnormal shapes and obliteration in comparison to the other groups. (**A**) Control, (**B**) kidneys from rats injected with saline solution, (**C**) kidneys from rats that received serum from healthy subjects and (**D**) kidneys from rats that received serum from patients having collapsing glomerulopathy. Abbreviations: EC = endothelial cell, *p* = podocytes, US = urinary space, BM = basal membrane.

**Table 1 biomedicines-08-00388-t001:** General characteristics of patients with collapsing glomerulopathy.

	Male	Female
Gender	4	2
Age (years)	29 ± 4	30 ± 11
Blood pressure (mmHg)	144/96 ± 9/9	143/91 ± 7/4
Blood urea nitrogen (mg/dL)	35.7 ± 29.9	18.5 ± 7.7
Creatinine clearance (mL/min)	55.8 ± 43.4	41.0 ± 36.7
Urinary protein (g/day)	7.0 ± 4.2	7.15 ± 3.4
Cholesterol (mg/dL)	288.0 ± 232.1	420.5 ± 3.5
Triglycerides (mg/dL)	254 ± 62	360 ± 188
Uric acid (mg/dL)	6.0 ± 1.1	5.5 ± 0.1
Analyzed glomeruli	17 ± 3	17 ± 2
Collapsed glomeruli	15 ± 5	17 ± 4

**Table 2 biomedicines-08-00388-t002:** Fatty acid composition of the serum from healthy subjects and collapsing glomerulopathy (CG) patients.

Fatty Acid	HS (*n* = 6)	CG (*n* = 6)
C16:0	18.51 ± 3.11	**23.07 ± 1.63 ****
C16:1n-7	1.45 ± 0.79	2.11 ± 1.34
C18:0	8.23 ± 1.06	8.11 ± 0.99
C18:1n-9	21.99 ± 1.79	24.97 ± 3.26
C18:2n-6	31.61 ± 2.22	30.32 ± 5.04
γ-C18:3n-6	0.50 ± 0.39	**0.07 ± 0.06 ***
α-C18:3n-3	2.00 ± 1.23	1.38 ± 0.58
C20:3n-6	1.29 ± 0.27	0.83 ± 0.62
C20:4n-6	7.04 ± 1.06	**4.33 ± 0.72 *****
C20:5n-3	4.73 ± 1.50	3.83 ± 1.70
C22:6n-3	3.29 ± 1.81	**0.95 ± 0.69 ****
SFA	26.74 ± 2.51	**31.19 ± 3.61 *****
MUFA	23.44 ± 2.25	27.09 ± 3.66
PUFA (n-3)	10.03 ± 3.03	**6.17 ± 1.75 ***
PUFA (n-6)	40.46 ± 2.75	35.56 ± 5.60

******p* = 0.04, ******
*p* = 0.01, and *******
*p* = 0.001, significantly different to CG vs. C. Data represent mean ± SD. Fatty acid nomenclature: C16:0, palmitic acid; C16:1n-7, palmitoleic acid; C18:0, stearic acid; C18:1n-9, oleic acid; C18:2n-6, linoleic acid; γ-C18:3n-6, γ-linolenic acid; α-C18:3n-3, α-linolenic acid; C20:3n-6, dihomo-γ-linolenic acid; C20:4n-6, arachidonic acid; C20:5n-3, eicosapentaenoic acid; C22:6n-3, docosahexaenoic acid. Abbreviations: SFA = saturated fatty acids, MUFA = monounsaturated fatty acids, PUFA = polyunsaturated fatty acids, HS = healthy subjects, CG = collapsing glomerulopathy.

**Table 3 biomedicines-08-00388-t003:** General characteristics in the experimental group.

	C	SS	HS	CG
Creatinine clearance (mL/min)	1.2 ± 0.1	1.0 ± 0.3	0.9 ± 0.1	**0.3 ± 0.1** *
Albuminuria (mg/to 120 h)	6.9 ± 2.8	14.6 ± 2.3	17.5 ± 3.8	**102.70 ± 11.9** *
Systolic blood pressure (mmHg)	115.4 ± 1.6	116.7 ± 2.3	114.4 ± 2.2	**140.1 ± 3.2** *
Kidney cholesterol (μg/mg of protein)	16.1 ± 6.1	14.5 ± 5.7	16.7 ± 5.6	20.3 ± 5.1
Serum cholesterol (mmol/L)	1.5 ± 0.2	1.6 ± 0.2	1.3 ± 0.2	1.6 ± 0.4

* C, SS and HS vs. CG *p* < 0.001. Values are mean ± SD (*n* = 6). Abbreviations: C = control, SS = saline solution, HS = healthy subjects, GC = collapsing glomerulopathy.

**Table 4 biomedicines-08-00388-t004:** Fatty acid compositions in the kidney homogenates from the experimental rats.

Fatty Acid	C	SS	HS	CG
C16:0	**25.8 ± 2.7** ^§^	**25.3 ± 3.7** ^†^	27.7 ± 0.7	**29.4 ± 0.8** **
C16:1n-7	0.7 ± 0.3 ^§§^	0.6 ± 0.3 ^†^	0.7 ± 0.4	1.2 ± 0.2 *
C18:0	16.8 ± 1.50	15.4 ± 1.3	16.8 ± 1.1	16.6 ± 1.4
C18:1n-9	17.1 ± 2.3	17.3 ± 2.3	16.6 ± 2.2	18.4 ± 1.3
C18:2n-6	12.8 ± 2.7	12.4 ± 1.6	12.6 ± 1.2	11.6 ± 1.5
γ-C18:3n-6	0.4 ± 0.1	0.7 ± 0.2	1.0 ± 0.9	0.4 ± 0.2
α-C18:3n-3	**0.5 ± 0.1** ^§^	0.3 ± 0.3	0.3 ± 0.2	0.2 ± 0.1
C20:3n-6	**0.7 ± 0.2** ^§^	**1.1 ± 0.5** ^†^	0.9 ± 0.1	**0.5 ± 0.2** ***
C20:4n-6	**18.9 ± 0.8** ^§§§^	**18.9 ± 0.4** ^††^	21.3 ± 1.0	**16.2 ± 0.8** ***
C20:5n-3	1.29 ± 0.36	1.9 ± 0.5	1.4 ± 0.4	1.2 ± 5.3
C22:6n-3	4.7 ± 0.9	5.3 ± 1.2	4.33 ± 0.84	4.3 ± 0.6
SFA	**42.7 ± 2.5** ^§^	**40.4 ± 2.9** ^†^	40.33 ± 0.84	**46.1 ± 1.2** *
MUFA	17.9 ± 2.5	**18.4 ± 1.9** ^†^	17.36 ± 2.22	**19.6 ± 1.1** *
PUFA (n-6)	**32.8 ± 1.9** ^§§^	**33.4 ± 1.6** ^††^	35.83 ± 1.53	**28.7 ± 0.9** ***
PUFA (n-3)	6.6 ± 1.1	7.5 ± 1.4	6.07 ± 1.20	5.8 ± 0.9

Data represent the weights of each individual FA/weights of total FAs as percentages (mean ± SD, *n* = 6 different animals). ^§^
*p* = 0.04, ^§§^
*p* = 0.01, and ^§§§^
*p* = 0.001, significantly different for C vs. CG; ^†^
*p* = 0.04 and ^††^
*p* = 0.01, significantly different for SS vs. CG; *****
*p* = 0.04, ******
*p* = 0.01 and *******
*p* = 0.005, significantly different for C vs. CG. Fatty acid nomenclature: C16:0, palmitic acid; C16:1n-7, palmitoleic acid; C18:0, stearic acid; C18:1n-9, oleic acid; C18:2n-6, linoleic acid; γ-C18:3n-6, γ-linolenic acid; α-C18:3n-3, α-linolenic acid; C20:3n-6, dihomo-γ-linolenic acid; C20:4n-6, arachidonic acid; C20:5n-3 eicosapentaenoic acid; C22:6n-3, docosahexaenoic acid. Abbreviations: C = control, SS = saline solution, HS = healthy subjects, GC = collapsing glomerulopathy, SFA = saturated fatty acid, MUFA = monounsaturated fatty acid, PUFA = polyunsaturated fatty acid.

**Table 5 biomedicines-08-00388-t005:** Indirect desaturation indexes of the desaturases in the kidney homogenates of the experimental rat groups.

Fatty Acid	C	SS	HS	CG
C16:1n-7/C16:0 (∆^9^)	**0.02 ± 0.01** ^§^	**0.02 ± 0.01** ^†^	0.03 ± 0.01	**0.04 ± 0.00** *
C18:1n-9/C18:0 (∆^9^)	1.02 ± 0.19	1.16 ± 0.19	1.03 ± 0.18	1.12 ± 0.17
C18:3n-6/C18:2n-6 (∆^6^)	0.035 ± 0.01	0.06 ± 0.02	0.08 ± 0.08	0.03 ± 0.01
C20:4n-6/C20:3n-6 (∆^5^)	**28.98 ± 9.61** ^§§^	**21.52 ± 11.35** ^††^	23.45 ± 6.43	**40.16 ± 19.12** **
C22:6n-3/C20:5n-3 (∆^5^)	3.82 ± 1.09	3.05 ± 1.25	3.23 ± 0.88	4.07 ± 1.42

The indirect desaturation index is obtained of the quotient of product on the substrate of fatty acid Data represent mean ± SD, (*n* = 6). ^§^
*p* = 0.04 and ^§§^
*p* = 0.01, significantly different for C vs. CG; ^†^
*p* = 0.05 and ^††^
*p* = 0.01, significantly different for SS vs. CG; *****
*p* = 0.04 and ******
*p* = 0.01, significantly different for HS vs. CG. Abbreviations: C = control, SS = saline solution, HS = healthy subjects, CG = collapsing glomerulopathy, ∆ = desaturase.

**Table 6 biomedicines-08-00388-t006:** Fatty acid composition in the serum of the experimental rat groups.

Fatty Acid	C	SS	HS	CG
C16:0	23.92 ± 1.53	22.04 ± 0.96	22.77 ± 1.39	**26.67 ± 1.71** **
C16:1n-7	**0.91 ± 0.16** ^§^	1.06 ± 0.02	0.84 ± 0.14	1.30 ± 0.15
C18:0	18.66 ± 0.38	19.62 ± 1.91	20.64 ± 2.23	18.39 ± 1.36
C18:1n-9	**14.21 ± 1.42** ^§^	**14.21 ± 0.87** ^†^	14.17 ± 0.84	**16.68 ± 1.47** ***
C18:2n-6	**16.41 ± 1.19** ^§^	**16.97 ± 2.98** ^†^	14.24 ± 2.56	12.12 ± 0.54
γ-C18:3n-6	**4.03 ± 0.77** ^§§^	**3.44 ± 1.24** ^††^	2.49 ± 0.80	**1.57 ± 1.01** *
α-C18:3n-3	1.2 ± 1.57	0.58 ± 0.26	2.08 ± 1.26	**0.29 ± 0.11** ***
C20:3n-6	0.48 ± 0.10 ^§^	0.40 ± 0.05 ^††^	0.40 ± 0.10	0.27 ± 0.05 *
C20:4n-6	15.221 ± 1.97	14.32 ± 1.37	17.80 ± 4.30	16.84 ± 1.59
C20:5n-3	1.51 ± 0.42	1.33 ± 0.28	2.02 ± 0.58	1.46 ± 0.74
C22:6n-3	4.44 ± 0.71	5.58 ± 2.15	4.65 ± 0.85	4.35 ± 0.71
SFA	42.58 ± 2.84	40.67 ± 1.91	43.41 ± 2.70	45.07 ± 1.44
MUFA	**15.12 ± 1.43** ^§^	16.27 ± 1.91	15.02 ± 0.85	**17.99 ± 0.37** ***
PUFA (n-6)	36.14 ± 2.64	35.14 ± 2.56	34.95 ± 2.82	30.81 ± 1.54
PUFA (n-3)	5.9 ± 0.65	6.92 ± 2.12	6.68 ± 1.40	5.81 ± 1.20

^§^*p* = 0.05 and ^§§^
*p* = 0.01, significantly different for C vs. CG; ^†^
*p* = 0.04 and ^††^
*p* = 0.01, significantly different for SS vs. CG; *****
*p* = 0.05, ******
*p* = 0.01 and *******
*p* = 0.001, significantly different for HS vs. CG. Data represent the weights of each individual FA/weights of total FAs as percentages (mean ± SD, *n* = 6 different animals). Fatty acid nomenclature: C16:0, palmitic acid; C16:1n-7, palmitoleic acid; C18:0, stearic acid; C18:1n-9, oleic acid; C18:2n-6, linoleic acid; γ-C18:3n-6, γ-linolenic acid; α-C18:3n-3, α-linolenic acid; C20:3n-6, dihomo-γ-linolenic acid; C20:4n-6, arachidonic acid; C20:5n-3 eicosapentaenoic acid; C22:6n-3, docosahexaenoic acid. Abbreviations: SFA = saturated fatty acid, MUFA = monounsaturated fatty acids, PUFA = polyunsaturated fatty acids.

**Table 7 biomedicines-08-00388-t007:** Indirect desaturation indexes of the desaturases in the fatty acids from serum of rats.

Fatty Acid	C	SS	HS	CG
C16:1n-7/C16:0 (∆^9^)	**0.038 ± 0.004** ^§^	0.05 ± 0.001	0.036 ± 0.005	**0.05 ± 0.009** *
C18:1n-9/C18:0 (∆^9^)	**0.76 ± 0.07** ^§^	0.82 ± 0.10	0.68 ± 0.089	**0.91 ± 0.12** **
C18:3n-6/C18:2n-6 (∆^6^)	**0.24 ± 0.06** ^§^	**0.21 ± 0.10** ^†^	0.17 ± 0.06	**0.12 ± 0.07** *
C20:4n-6/C20:3n-6 (∆^5^)	**33.30 ± 11.46** ^§^	**35.91 ± 5.09** ^††^	27.28 ± 11.51	**64.33 ± 13.84** **
C22:6n-3/C20:5n-3 (∆^5^)	3.19 ± 1.24	4.35 ± 1.74	2.39 ± 0.38	3.57 ± 1.55

Data represent the quotients of the percentages of each individual FA (product)/percentages of their substrates (mean ± SD, *n* = 6). ^§^
*p* = 0.03, significantly different for C vs. CG; ^†^
*p* = 0.04 and ^††^
*p* = 0.01, significantly different for SS vs. CG; * *p* = 0.05 and ** *p* = 0.01, significantly different for HS vs. CG. Abbreviations: C = control, SS = saline solution, HS = healthy subjects, CG = collapsing glomerulopathy.

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
