# Peer review of "Alteration of the Fatty Acid Metabolism in the Rat Kidney Caused by the Injection of Serum from Patients with Collapsing Glomerulopathy"

_biomedicines, 2020, doi:10.3390/biomedicines8100388_

Round 1

Reviewer 1 Report

Major Comments

The above manuscript is well – written and comprehensive. Authors injected serum from patients suffering from collapsing FSGS to rats and they induced collapsing FSGS changes in glomeruli and alterations in the fatty acid metabolism, suggesting that these changes play a role in the elevation of systolic blood pressure and renal injury. Moreover, they combined different methods in their research and they visualized their results, by light, electron microscopy and immunohistochemistry, using in parallel control groups. Discussion is well documented and literature is sufficient.

However, limitation of the study is the absence of a group from patients suffering from proteinuria having other pathologies than collapsing FSGS, whom serum could also be injected in rats. Such serum can be used as control group, in order to confirm that soluble factors of collapsing FSGS patients, as well lipid metabolism deterioration is different (or it is not), from other pathologies causing proteinuria, or even to determine if other factors play also a role. Since a certain factor causing collapsing FSGS is not yet established, this control addition would be necessary for clarifying pathogenesis. Maybe, serum from patients with classical FSGS, could be used as the most appropriate control group, in order to investigate if lipid status is changed.

Furthermore, authors state that proteinuria is progressing through the treatment. However, they could also investigate, whether proteinuria resolves after the experiment takes place, or if feasible, how long it takes to resolve (and if there is any resolution at all). Another important observation would be, if there is residual structural damage in podocytes and glomeruli after stopping injections. Authors could address this issue, by adding a new group in their research.

Glomerular “collapse” with the classic view, is not demonstrated in the images of light and electron microscope. Instead, podocyte hypertrophy, podocyte and parietal cell prominence and activation, as well as reduced glomerular size and glomerular retraction are demonstrated and possibly are the early (and possibly only) findings. Likely, these changes are a “precursor” of collapsing FSGS, but still, glomerular loops collapse is not clearly demonstrated (probably, true collapse needs more time to be settled). This observation should be emphasized in the paper and correction is needed when necessary (including light and electron microscope legends). Also a comment for the rest of renal parenchyma should be added (especially if tubular dilation with proteinaceous casts was noted).

Minor Comments

In the Figure 2d, page 8, podocyte hypertrophy is difficult to be appreciated. Maybe another image could be more suitable to demonstrate podocyte hypertrophy and activation.

In figure 5, page 11, line 287, probably authors would like to say “negative for visceral cells” instead of “negative for collapsing”. This type error must be corrected.

Author Response

Question 1

However, limitation of the study is the absence of a group from patients suffering from proteinuria having other pathologies than collapsing FSGS, whom serum could also be injected in rats. Such serum can be used as control group, in order to confirm that soluble factors of collapsing FSGS patients, as well lipid metabolism deterioration is different (or it is not), from other pathologies causing proteinuria, or even to determine if other factors play also a role. Since a certain factor causing collapsing FSGS is not yet established, this control addition would be necessary for clarifying pathogenesis. Maybe, serum from patients with classical FSGS, could be used as the most appropriate control group, in order to investigate if lipid status is changed.

Answer

The purpose of this study was to investigate the mechanism by which the intravenous administration of serum from CG patients into rats induces high systolic blood pressure and modifies the FA profile in the kidney and serum in rats. The study was not designed to confirm if a soluble factor only present in the serum of patients with the collapsing form of FSGS is the cause of the changes in blood pressure and FA profile in rats. The study proposed by the reviewer is certainly of importance but our study was not aimed to clarify this issue. Furthermore, it is not possible to add the results from a new experimental group and change the aim of the study in the 10 days we were given to reply the suggestions made by the referees. Also in 2004, Avila-Casado et al. show that the injected serum of the  FSGS and CG in the same rat model present the glomerular damage in CG in comparison with FSGS. (Avila-Casado, M.C.; Perez-Torres, I.; Auron, A.; Soto, V.; Fortoul, T.I.; Herrera-Acosta, J. Proteinuria in rats induced by serum from patients with collapsing glomerulopathy. Kidney. Int. 2004, 66, 133–143, doi: 10.1111/j.1523-1755.2004.00715. x.)

Question 2

Furthermore, authors state that proteinuria is progressing through the treatment. However, they could also investigate, whether proteinuria resolves after the experiment takes place, or if feasible, how long it takes to resolve (and if there is any resolution at all). Another important observation would be, if there is residual structural damage in podocytes and glomeruli after stopping injections. Authors could address this issue, by adding a new group in their research.

Answer

The issues raised by the reviewer is of importance but we would need to repeat the experimental protocol to solve the question and this cannot be done in the 10 days we are given to reply the suggestions made by the reviewers.

Question 3

Glomerular “collapse” with the classic view, is not demonstrated in the images of light and electron microscope. Instead, podocyte hypertrophy, podocyte and parietal cell prominence and activation, as well as reduced glomerular size and glomerular retraction are demonstrated and possibly are the early (and possibly only) findings. Likely, these changes are a “precursor” of collapsing FSGS, but still, glomerular loops collapse is not clearly demonstrated (probably, true collapse needs more time to be settled). This observation should be emphasized in the paper and correction is needed when necessary (including light and electron microscope legends). Also a comment for the rest of renal parenchyma should be added (especially if tubular dilation with proteinaceous casts was noted).

Answer

Thanks you for your comment. if, it is correct, now we have corrected this observation and added the suggestion in the document.

Minor Comments

Question 4

In the Figure 2d, page 8, podocyte hypertrophy is difficult to be appreciated. Maybe another image could be more suitable to demonstrate podocyte hypertrophy and activation.

Answer:

Done, the figure was replaced

Question 5

In figure 5, page 11, line 287, probably authors would like to say “negative for visceral cells” instead of “negative for collapsing”. This type error must be corrected.

Answer

Done, the phrase was replaced

Reviewer 2 Report

In this manuscript Soria-Castro and colleagues investigate the changes in renal fatty acid metabolism in rats injected with serum from patients with collapsing glomerulopathy. Although interesting, the manuscript is very descriptive, lacks mechanistical insight and is limited in scope.  

Comments:

  1. The manuscript would benefit from language editing in order to eliminate spelling errors and improve readability. Especially the Discussion section can be improved regarding clarity.

  1. Figure 1 basically shows the same data as Table 1. This is redundant. Please show the data only in a figure or as a table.

  1. The graphs showing densitometric analysis seem not to be included within the figures, are not properly described in the figure legend and do not show all the statistically significant differences that are described in the text. Moreover, it would be better to show the data as dot plots.  

  1. Statistics shown in most tables is unclear. The legend states “significantly different from CG vs. C, SS and HS. Followed by a statement as “***p=0.001”. I find it difficult to believe that the p value of three comparisons is exactly 0.001 for all.

  1. Several IHC pictures are of poor quality, and all figures should include a lower magnification per group.

  1. Overall, the manuscript is very descriptive, and a lot of data is a characterization of the model instead of focusing on FA metabolism. Moreover, the manuscript fails to provide insight into a potential mechanism. I believe that the manuscript would greatly benefit from more data on inflammation and prostaglandins.

Author Response

Question 1

The manuscript would benefit from language editing in order to eliminate spelling errors and improve readability. Especially the Discussion section can be improved regarding clarity.

Answer

Done, the manuscript was reviewed by a native speaker in the English language and the discussion section was improved giving metabolic pathways that could participate in collapsing glomerulopathy.

Question 2

Figure 1 basically shows the same data as Table 1. This is redundant. Please show the data only in a figure or as a table.

Answer

Done, the table 1 was deleted

Question 3

The graphs showing densitometric analysis seem not to be included within the figures, are not properly described in the figure legend and do not show all the statistically significant differences that are described in the text. Moreover, it would be better to show the data as dot plots.

Answer

Done, the writing of these graphs was improved, the missing statistical differences were also added, and the graphs were replaced by dot plots

Question 4

Statistics shown in most tables is unclear. The legend states “significantly different from CG vs. C, SS and HS. Followed by a statement as “***p=0.001”. I find it difficult to believe that the p value of three comparisons is exactly 0.001 for all.

Answer

The statistical values of the comparison between groups were added throughout the results section and in the tables and graphs

Question 5

Several IHC pictures are of poor quality, and all figures should include a lower magnification per group.

Answer

The images with low resolution of IHC were changed for others with higher resolution; we also added a representative photo at lower magnification (200x) only with PAS staining, since it is the most specific technique to demonstrate changes at the tubular level.

Question 6

Overall, the manuscript is very descriptive, and a lot of data is a characterization of the model instead of focusing on FA metabolism. Moreover, the manuscript fails to provide insight into a potential mechanism. I believe that the manuscript would greatly benefit from more data on inflammation and prostaglandins.

Answer

In this new version we have added in the discussion section, a possible participation of the fatty acid pathways that could be involved in collapsing glomerulopathy.

We also egregate the immune histochemine of 5-lipoxygenase, an enzyme that metabolizes arachidonic acid in HPETE and HEETEs, which participates in the process of vasoconstriction and inflammation.

We only add this immunostaining since it is with the antibody that we currently have in our laboratory

Round 2

Reviewer 1 Report

Comments

The corrections made on the histology part are documented and sufficient.

Authors answered the questions and responded to the suggestions made, as much as they could, given the time - period they had to respond.

My only observation is that authors should add an explanation in the text of the Discussion section, why they selected serum from collapsing FSGS patients to induce alterations on lipid metabolism of rats and exclude serum from patients with other “proteinuric” pathologies, such as classical FSGS. In any case, they should give their assumption/hypothesis to the readers, if there is any correlation between modifications of lipid profile and the certain disease they chose to investigate, as they did to their response in my comments. If, only serum from collapsing FSGS patients (and not serum from patients with other proteinuric diseases), induce these changes to the kidneys and lipid profile, readers should be aware of it. On the other hand, if there is no a definitive answer, readers should also know and possibly reveals a limitation of the study. I believe a comment from authors in the text of the Discussion section would be appropriate, in order to justify their, otherwise, sufficient research to the readers.  

Author Response

We fully appreciate the time spent by the reviewers in evaluating our paper as well as their comments and advice.

I am sending the new version of the revised paper Biomedicines-912410 Article title: ALTERATION IN THE FATTY ACIDS METABOLISM IN THE KIDNEY CAUSED BY THE INJECTION OF SERUM FROM PATIENTS WITH COLLAPSING GLOMERULOPATHY by authors: Elizabeth Soria-Castro, Veronica Guarner-Lans, María Elena Soto,Carmen Avila, Linaloe Manzano Pech, Israel Pérez-Torres. We have tried to follow their suggestions and enclose our replies, which are marked in red.

Best regards

Israel Pérez-Torres PhD.

Referee 1

Comments

Question 2

English language and style are fine/minor spell check required

Answer

Done, the English language was revised

Question 2

My only observation is that authors should add an explanation in the text of the Discussion section, why they selected serum from collapsing FSGS patients to induce alterations on lipid metabolism of rats and exclude serum from patients with other “proteinuric” pathologies, such as classical FSGS. In any case, they should give their assumption/hypothesis to the readers, if there is any correlation between modifications of lipid profile and the certain disease they chose to investigate, as they did to their response in my comments. If, only serum from collapsing FSGS patients (and not serum from patients with other proteinuric diseases), induce these changes to the kidneys and lipid profile, readers should be aware of it. On the other hand, if there is no a definitive answer, readers should also know and possibly reveals a limitation of the study. I believe a comment from authors in the text of the Discussion section would be appropriate, in order to justify their, otherwise, sufficient research to the readers.

Answer

Done, a section stating the limitations of the study was added in final part of the document.

Referee 2

Question 1

English language and style are fine/minor spell check required

Answer

Done, the English language was revised

Reviewer 2 Report

The authors have adequately addressed my comments.

Author Response

(The authors gave the same response as above.)
